# An album is a story: Feature arcs in sequences of tracks

**Pedro Neto** [ID]◉*, **Martin Hartmann** [ID]◉, **Geoff Luck**◉, **Petri Toiviainen** [ID]◉

Center of Excellence in Music, Mind, Body and Brain. Department of Music, Arts and Culture Studies - University of Jyväskylä, Jyväskylä, Finland

◉ These authors contributed equally to this work.
* pdealcan@jyu.fi

**Data availability statement:** All relevant data are within the paper and its Supporting Information files or from https://osf.io/69n3x/.

## Abstract

When releasing an album, musicians often think about the order in which their songs will be presented to the listener. In a previous study, we analyzed sequencing patterns within a set of 50,000 published works and found that louder, more energetic tracks are usually placed before quieter ones. Here we asked 130 music professionals to deliberately arrange the order of songs for a hypothetical album release. Our findings reveal consistent statistical patterns of track sequencing, with musicians showing broad agreement on the placement of songs within an album. Overall, tempo follows an inverted U-shape, while valence, arousal and loudness exhibit the opposite pattern. Results are interpreted in light of auditory science, and compared with sequencing trends commonly found in song lyrics and literary works.

## 1 Introduction

Human activities are often performed according to a set of well established sequencing principles. These principles can dictate, for instance, the order in which ingredients should be added to a dish [1], the progression of exercises in a gym routine [2], and the types of quests best suited for the early, middle and late stages of a game [3]. In every case, these principles are intended to facilitate, improve, or even enable the success of the activity that is being performed.

As a time-based activity, music is often seen as the art of organizing sounds into a sequence. French composer André Hodeir notes that "a musical phrase, no matter how beautiful it is, reaches its expressive summit only when it is in perfect harmony with preceding and following phrases" (as cited in [4, p. 811]). Contemporary artists often agree, and state that the order in which tracks are presented in an album can affect the musical experience of the listener [5–8].

Anecdotally, musicians hold varying beliefs about how songs should be sequenced. It has been said, for instance, that pairs of consecutive tracks should not be too similar in terms of tempo and key [6–8], but that similar tonalities are acceptable if the paces of the songs are dramatically different [5]. According to ref. [6], musicians should not put two slow songs next to each other, and singer David Brewis adds that "if you go straight from a quite fast song [...]

**Funding:** This research was funded by the Research Council of Finland, under the Center of Excellence in Music, Mind, Body and Brain (Project number: 346210).

**Competing interests:** The authors have declared that no competing interests exist.

to just a little bit of a slower song, it can make the slower song seem like it's dragging. We have to avoid that" [5].

Musicians also seem to pay special attention to the first song of an album [7,9–11]. According to [11], "if you don't catch people right off the bat, they might not hear the hits at the end." In an interview to Garvey [5], producers Peter Hammill and Ashley Abram corroborate this idea, suggesting that "hits" should be at the front of the album.

It is also believed that album sequences should establish a well-defined trajectory from the first to the last track. According to [10], "arranging songs by key from lowest to highest creates a positive feeling, and a gradual increase in tempo through an album might evoke a rising sensation." Hahn [9] suggests, in addition, that if sequences of tracks "build and release tension over the whole release [...] tracks will hit harder individually, and the overall effect [of the album] will be enhanced."

Inspired by these anecdotal beliefs, as well as by a lack of systematic investigations on this issue, we conducted a study [12] in which track features were analyzed for 51010 published albums. It was found that songs with high values of valence, energy and loudness are more likely to be positioned at the beginning of each album, and that consecutive tracks tend to alternate between increases and decreases of valence and energy. Overall, our findings expose the algorithmic nature of album sequencing, and suggest that some rationales are followed by a wide range of artists and producers, somewhat independent from musical genre (Fig 1).

Albums were chosen for our previous investigation because these are prime examples of a musical context in which professionals make deliberate decisions about the order in which

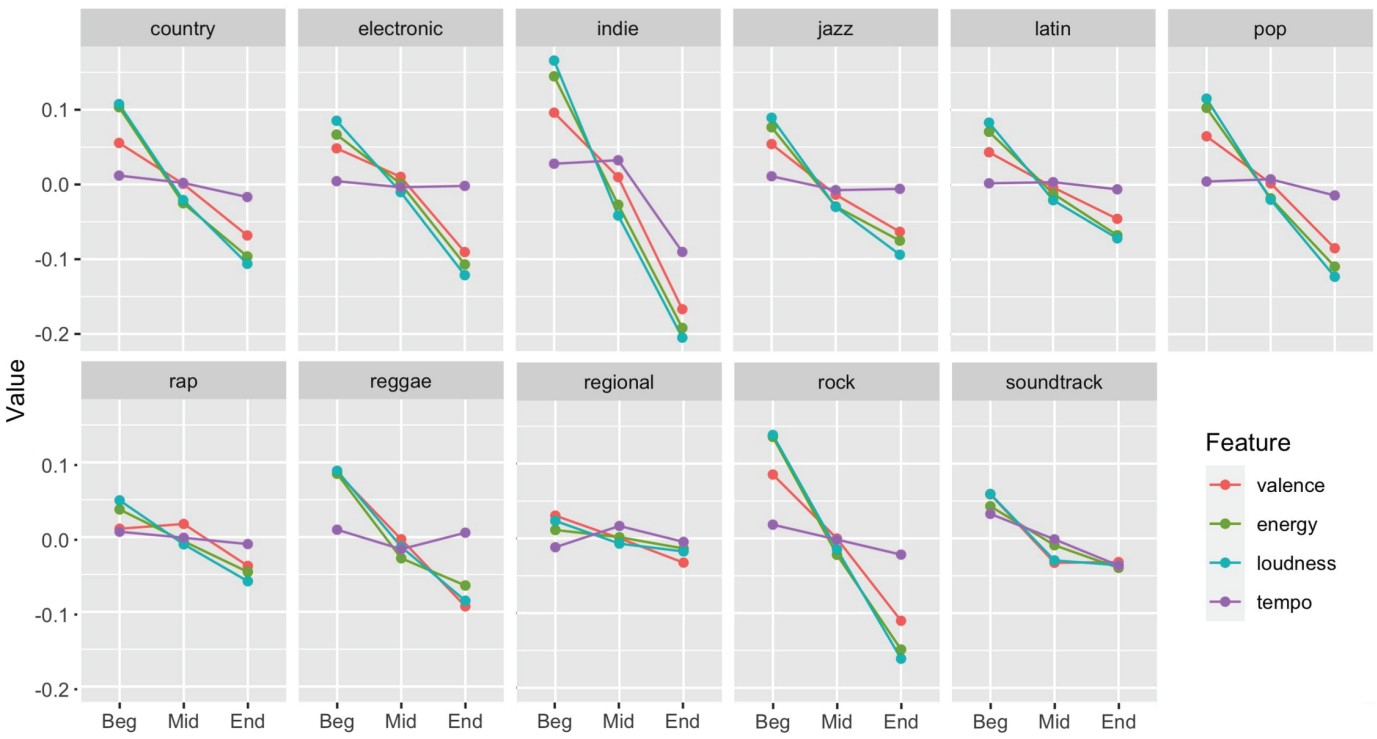

**Fig 1. Feature values along album segments.** Standardized feature distribution of 51000 albums found by [12]. Given a varying number of tracks, albums were discretized into 3 segments (i.e., beginning, middle, end). Feature values represent the average feature for all tracks pertaining to a given album segment.

their tracks will be presented to the listener. While playlists available on major streaming platforms may also be curated with attention to track order, these are essentially different from albums in a few key aspects: they are not necessarily created by music professionals, they can be shuffled and modified by any user, and they may serve a diverse range of purposes, such as exercise, mood regulation and background listening [13–15].

Even though we chose albums as a test case for our track sequencing investigation, findings in this area have the potential to fuel the development of applications in other musical contexts, such as automated playlist generation [16], DJ set creation [17], and music recommendation algorithms [18], which can incorporate principles used by professional musicians to determine the order in which their songs will be presented to the listener.

In our previous study, our analyses were limited to a kind of between-subjects exploration of album sequencing, where each musician/music producer sequenced a specific set of tracks for their own album. We could not investigate, for instance, the extent to which musicians agree about the ordering of a common set of songs. Here, we focus on expanding upon our previous results by asking professional musicians to sequence a set of tracks, as if they were to be released on one of their albums. We also improve on the reproducibility and interpretability of our last study by computing audio features with the open-source MIRtoolbox [19], rather than using proprietary features computed by Spotify. Finally, we administer an exploratory survey to investigate participants' opinions about different aspects of album sequencing.

## 2 Materials and methods

### 2.1 Participants

Professional musicians and music producers were recruited through social media channels, and no compensation was offered for their participation. Our final sample comprised 130 individuals who had played an average of 263.9 concerts (SD = 663.8), recorded 6.12 albums (SD = 15.11), and had 14.1 years of musical experience (SD = 10.9), for detailed participant statistics, see Supporting Information (S2. Participants' musical experience). A total of 25 participants answered the survey in Portuguese, whereas the other 105 in English. We did not exclude any participant based on their musical experience, and considered their self-report of professional musicianship both necessary and sufficient for participation in our study.

### 2.2 Stimuli

For the sequencing task, we devised 3 sets of 5 tracks within jazz and classical genres. These genres were chosen because, in comparison with Electronic Dance Music (EDM) and Funk, for instance, jazz and classical compositions are notorious for offering wider variation of rhythm, harmony, and dynamics from one piece to another. This is important because, if music-related features influence how albums are sequenced, as suggested by anecdotal evidence [5–8], as well as by our previous study [12], maximizing the variability of song styles within each set would make this effect clearer. Finally, Jazz and Classical music are often instrumental, which allows us to control for the possible effect of lyrics.

A relatively small number of pieces was chosen in order to minimize the necessary time to complete the online survey. Also, the choice of working with only 2 genres is supported by our previous study, which suggested that sequencing patterns are, at least to some extent, generalizable between different genres (Fig 1; [12]). Still, we chose to work with 3 distinct sets of songs, as a wider spectrum of features values may increase the generalizability of our results.

Set 1 was composed of pieces from a Jazz album with relatively low popularity (i.e., less than 1000 monthly listeners on Spotify), which controls for possible effects of familiarity [20]. This album also offered high recording quality, a wide spectrum of track styles, and a single instrumental formation throughout all tracks (i.e., piano and electric guitar), which would also control for the effect that instrumentation may have on track sequencing.

Tracks within sets 2 and 3 were manually selected from the Dataset on Emotional Annotations [21], which is composed of 400 pieces from the recording company Magnatune, equally divided between classical, rock, pop and electronic genres. As reported by [21], this dataset provides high quality audio recordings, and pieces that are generally not known by a wide audience. Only classical tracks were chosen for both sets because of their abundance of instrumental compositions. Individual tracks were selected irrespective of authorship or album.

For set 1, we manually selected 60-second segments that we judged to be representative of their respective tracks. We did not compare the features of these segments with those of the full tracks; rather, selection was based on perceptual judgment. While systematic audio thumbnailing approaches [22] could have been employed, we argue that the representativeness of specific segments is not particularly relevant to the research question posed in this study. Even if a segment is not fully representative of the entire track, it still allows us to analyze how its features influence sequencing decisions. Tracks in Sets 2 and 3 were pre-segmented into 60-second clips within the dataset provided by [21]. All clips included 1-second fade-ins and fade-outs. A list of song names and corresponding clips is available as Supporting Information, S1. Track information.

## 2.3 Procedures

Each participant was assigned to one of the 3 sets of songs. Their task was to organize the tracks into a sequence that would be released in a musical album. The computer interface presented to participants is displayed in Fig 2. The initial disposition of tracks was randomly determined to each participant. Songs could be heard any number of times, and there was no time limit for the completion of the task. After sequencing, participants were presented with a survey comprising the following propositions:

(a) Determining a good song sequence is important for the overall quality of an album
(b) In an album, there should not be 3 or more consecutive songs that are too similar to each other
(c) Usually, it is better for an album to start with the most striking and energetic songs
(d) Out of all the time I spend thinking about song ordering, I spend most of this time trying to decide which one will be the first song of my albums
(e) Usually, it is better for an album to end with the most striking and energetic songs
(f) In an album, there should not be 3 or more consecutive songs in the same tempo
(g) Out of all the time I spend thinking about song ordering, I spend most of this time trying to decide which one will be the last song of my albums
(h) In an album, there should not be 3 or more consecutive songs in the same tonality
(i) In an album, consecutive songs should be as different as possible from each other

Propositions were rated on a visual analog scale (Fig 2) from 0 to 100, with 0 representing "Completely disagree" and 100 representing "Completely agree."

We ran our survey and sequencing tasks in 3 separate stages, one for each set of songs. Recruitment occurred between 09.11.2021, and 05.02.2022. Responses for set 1 were collected over a period of 2 months, and responses for sets 2 and 3 were collected over a period of one

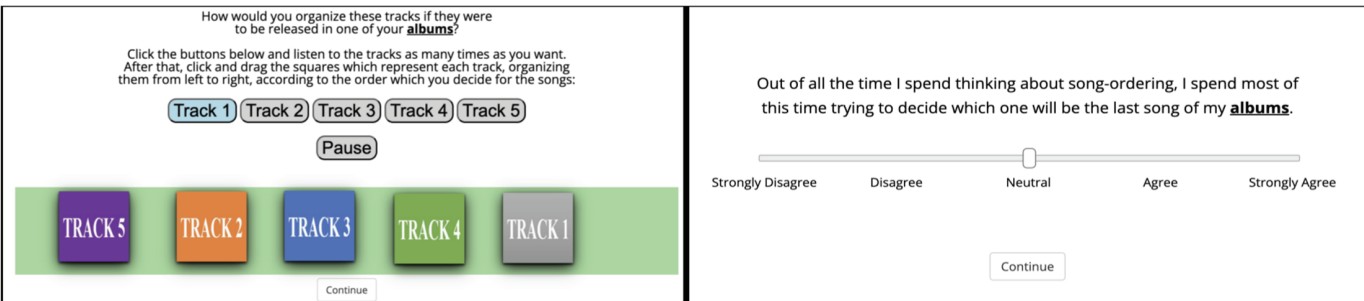

**Fig 2. Computer interface** Computer interface used to sequence tracks and rate survey propositions.

month each, concomitantly. Set 1 was sequenced by 62 participants, while sets 2 and 3 were sequenced by 37 and 31 participants, respectively.

All surveys were performed in accordance with the guidelines and regulations of the National Advisory Board on Research Ethics in Finland (TENK, see https://www.tenk.fi/sites/tenk.fi/files/ethicalprinciples.pdf). Individuals were informed about their rights as participants, the content of the research, and gave their informed consent by clicking an Agree button that appeared on the computer screen. Furthermore, all data was collected anonymously, and no sensitive information was obtained.

### 2.4 Sequencing analysis

Two aspects of track sequencing were analyzed: positional regularity (PR), and feature distribution. The first set of analysis indicated the extent to which participants agree about how to position specific tracks in albums, whereas the later indicated how different music-related features are distributed in a sequence of songs.

PR is defined as the normalized Shannon entropy $\eta$ of track frequencies within each album position. An album is composed by a set of 5 tracks $L = \{A, B, C, D, E\}$, and a list of positions $P = (P1, P2, P3, P4, P5)$. For each position $p \in P$, we define the frequency vector $\mathbf{f}_p = (f_p^A, f_p^B, f_p^C, f_p^D, f_p^E)$, where $f_p^l$ represents the frequency with which track $l \in L$ was chosen for position $p \in P$. The vector $f_{P1} = (0.6, 0.0, 0.0, 0.4, 0.0)$ indicates, for instance, that tracks A and D were chosen for $P1$ 60% and 40% of the times, respectively. $PR$ is thus calculated as:

$$\eta(\mathbf{f_p}) = 1 - \frac{H(\mathbf{f_p})}{H_{\max}} = 1 - \sum_{l \in L} \frac{f_p^l \log_2(f_p^l)}{\log_2(|f_p|)} \tag{1}$$

High PR indicates that one or more tracks are chosen for a given position with a higher probability than the others (e.g., 90% of participants choose track A as the opening of an album). Conversely, if PR is low, all tracks have approximately the same probability of being placed in a given album position (i.e., 20%).

For hypothesis testing, we created a control dataset consisting of 1,300,000 randomly generated track sequences. This number was obtained by shuffling each participant-generated album 10,000 times. Statistical significance was then calculated by means of bootstrapping, where we iteratively computed $PR_{sim}$ from a subsample of 62, 37 and 31 albums within sets 1, 2 and 3 of the control dataset, respectively. An empirical cumulative distribution (ECDF) was obtained by repeating this process 10,000 times, allowing for replacement. Finally, p-values

were calculated as the probability of obtaining a given $PR_{true}$ or higher in the ECDF of $PR_{sim}$, which represents the null hypothesis.

## 2.5 Feature analysis

For feature analyses, we computed tempo, loudness, valence, and arousal of each track, and evaluated how each feature was organized sequentially. Instead of focusing on lower-level features, such as spectral flux and zero-crossing rate, we chose to work with features that are easily interpretable in terms of their perceptual value.

Valence represents how positive or negative a given stimulus is perceived to be, while arousal measures the perceived intensity of the stimulus on a scale from low to high. In music, valence and arousal have been shown to have predictive and explanatory power with regards to memory [23,24], neural activity [25], and even decision making [26]. Furthermore, these dimensions of perceived emotion have been shown to be robust across cultures, leading authors to hypothesize that valence and arousal represent universal aspects of music perception [27].

In this study, valence and arousal are computed with a Multiple Linear Regression model proposed by [28]. This method is implemented as MIRtoolbox's miremotion function [19], and it estimates emotional ratings from lower level acoustic features such as spectral centroid, Root Mean Square (RMS), spectral spread, entropy, and spectral novelty. As reported by [28], this model was trained on human ratings of 360 soundtrack excerpts, given by 116 participants, and was capable of explaining 70% of the variance in their data. Predictions from miremotion have been used, for instance, in the context of emotion visualization systems [29], movie-scene emotion recognition [30], and modeling of bodily sensations [31].

Loudness and tempo, in turn, are also important musical features, which are not only used to calculate valence and arousal [19,32], but have themselves been shown to affect cognitive processing speed [33], music-induced chills [34] and music-induced movement [35,36], for instance. Loudness and tempo were also computed with MIRtoolbox version 1.3 [19].

In order to test the hypothesis that feature values would differ between positions of an album, we conducted a mixed-design Analysis of Variance (ANOVA) with position (i.e., positions 1 to 5) and feature (i.e., valence, arousal, loudness and tempo) as within-subjects factors, and set (i.e., 1 to 3) as between subjects factor. Feature values were converted to z-scores for each participant and feature.

In order to test if, in accordance with annecdotal beliefs (see Introduction), musicians follow some patterns of feature change between consecutive tracks, we propose the metric of overall feature dissimilarity $D$, which computes the Euclidean distance between audio features of tracks $i$ and $i + 1$. Let $t_i = (v_i, a_i, l_i, \tau_i)$ represent the feature vector of track $i$ where $v_i, a_i, l_i$ and $\tau_i$ are valence, arousal, loudness and tempo values, respectively. The index $j$ refers to the feature dimension, with $j = 1$ corresponding to valence, $j = 2$ to arousal, and so on. The average feature difference between a given pair of tracks is thus given by:

$$D_i = \frac{1}{4} \sum_{j=1}^{4} \| t_{i+1}^{(j)} - t_i^{(j)} \| \tag{2}$$

Similarly, tempo dissimilarity (survey proposition (f)) was computed as the Euclidean distance between consecutive tracks, but with $t_i$ composed only of tempo values.

Key dissimilarity (survey proposition (h)) between consecutive tracks was assessed with the function mirkeystrength from MIRtoolbox [19], which computes correlations between

the chromagram of a given audio segment and all key-profiles from [37]. The result is a vector of 24 key strength values, which corresponds to each one of the 12 tonalities within both major and minor modes. Key dissimilarity was then calculated by computing the Pearson correlation between key strength profiles of consecutive tracks within each album. We did not use Euclidean distance to calculate this metric because it would be sensitive to the magnitude of key-strength profiles, rather than only to its overall shape. For consistency with the other measures, we show results as $1 - \rho$, which converts key dissimilarity into key dissimilarity, in a range between 0 and 2. Statistical significance of key, tempo and overall feature dissimilarity was obtained with the same bootstrap approach presented in Section 3.5.

## 2.6 Survey analysis

Upon visual inspection of survey responses, we identified potential polarization of opinions between agree, disagree and neutral. This trend was confirmed through Hartigan's dip test (detailed statistics available as Supporting Information, S3. Detailed survey statistics), which measures how much a given distribution deviates from unimodality. In the context of our survey, departures from uni-modality cause central-tendency measures to render misleading interpretations.

For instance, take the hypothetical scenario where half of the participants completely agreed with proposition (a), while the other half completely disagreed. The resulting response distribution would be bimodal, and central tendency measures would suggest that participants are neutral regarding $A$, while, in fact, responses are strongly polarized. To address this issue, we discretized participant responses according to the following piecewise function $f$:

$$f(x) = \begin{cases} disagree & \text{if } x \leq 33 \\ neutral & \text{if } 33 < x < 66 \\ agree & \text{if } x \geq 66 \end{cases} \tag{3}$$

The frequency of each categorical response was computed and then subjected to a $\chi^2$ test, which does not test for the significance of a central tendency of opinions, but rather indicates if participants leaned towards one or more segments of the scale (i.e., disagree, neutral, agree). Finally, the extent to which responses are distributed throughout these segments is visualized with a density plot.

## 3 Results

### 3.1 Sequencing regularities

For most positions and sets, the frequency of tracks is visibly different from baseline levels (i.e., 20%), indicating that track sequencing was not performed randomly, as shown in Fig 3 (left-side). For instance, in set 2, track B was chosen for position 1 (P1) by 37% of the musicians, while track C, was chosen 0% of the times. Overall, large differences between the frequency of a given track and the baseline of 20% will lead to high PRs (Fig 3, right side).

The distribution of $PR_{simulated}$ across all sets had a mean and standard deviation of .0096 and .006, respectively (Fig 4, left-side). Fig 4 shows the $PR_{true}$ for each position in relation to the distribution of $PR_{simulated}$. As visible in Fig 3 and 4, highest PR was found for position 1 ($PR = .16$, $p<.001$), followed by position 3 ($PR = .09$, $p<.001$), position 4 ($PR = .05$, $p<.001$), position 5 ($PR = .05$, $p<.001$), and position 2 ($PR = 0.02$, $p<.001$).

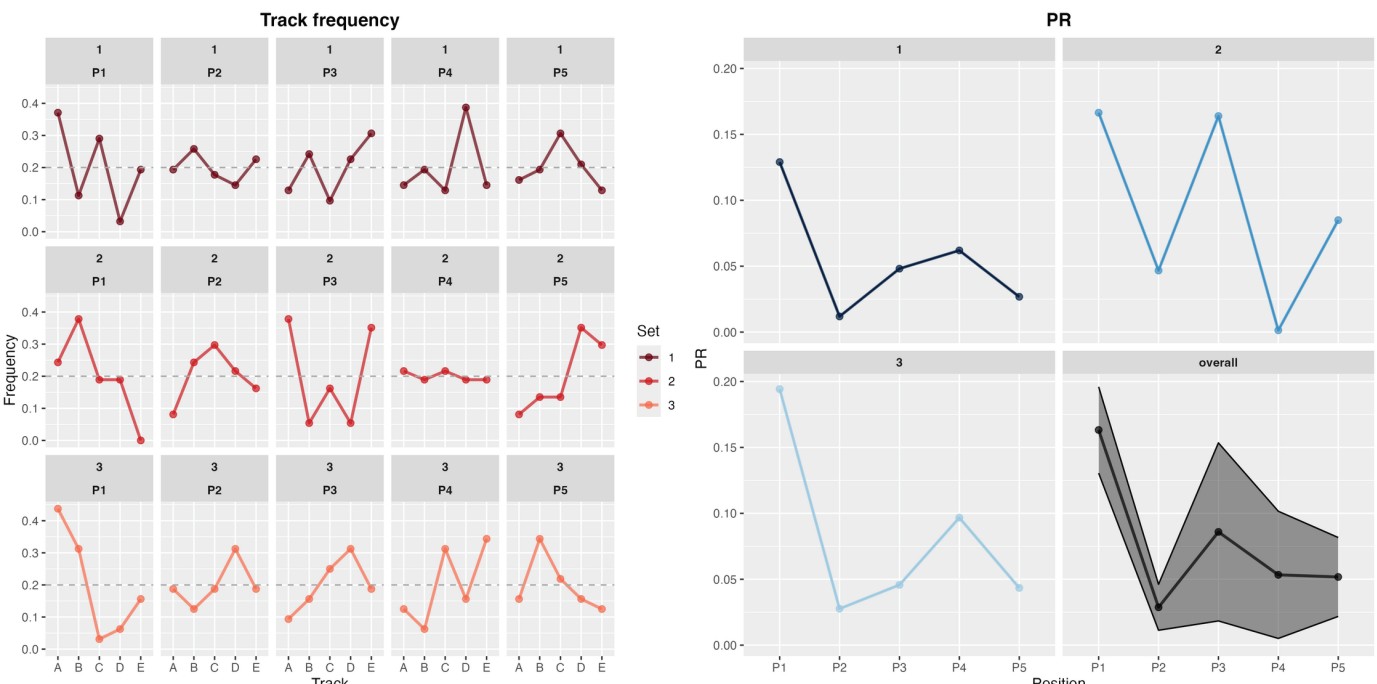

**Fig 3. Track frequency by position and set.** *PR* by position and set, as well as *PR* across all sets (overall). Shaded area represents the standard deviation for each position.

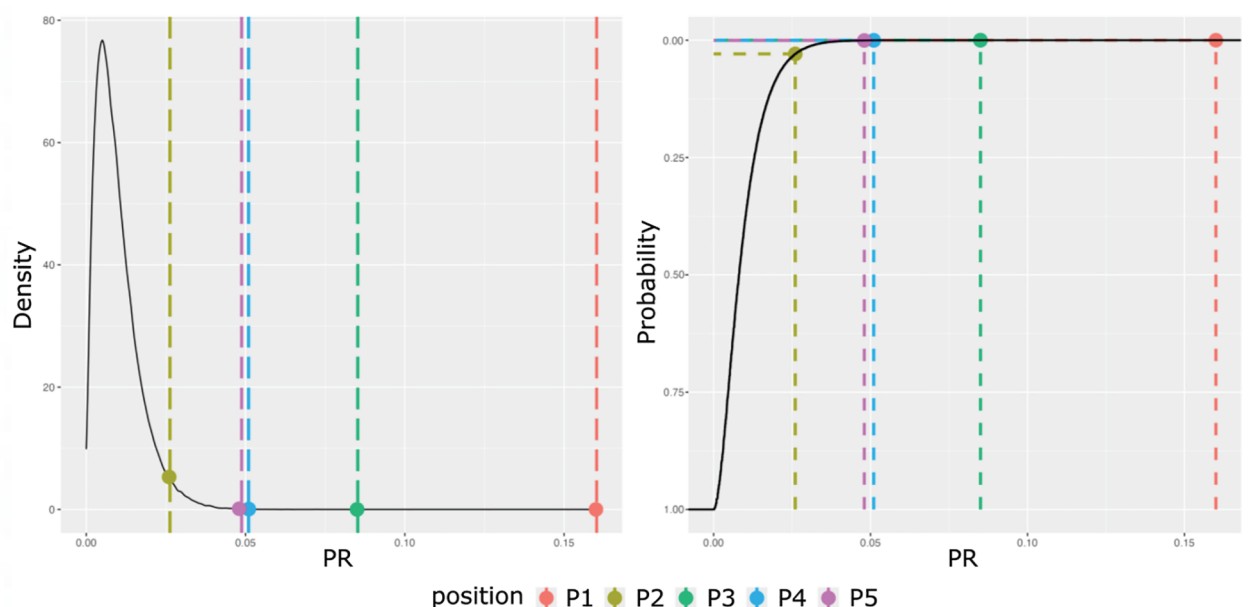

**Fig 4. Distributions of *PR*.** Left: density plot of $PR_{simulated}$ with respect to $PR_{true}$ obtained from participant-sequenced albums. Right: $PR_{true}$ (dashed lines) in relation to the Empirical Cumulative Distribution obtained from the control condition (solid line), where the y axis indicates the probability of obtaining a value equal to, or greater than a given *PR*.

## 3.2 Feature analysis

Results of the ANOVA indicated that there was a significant main effect of position, $F(3.81, 483.63) = 6.76, p < .001, \eta^2 = .018$, suggesting that feature values differed across positions of the album. There was also a significant interaction between position and feature $F(7.87, 999.39) = 2.80, p < .01, \eta^2 = .014$. Effect sizes were small for both effects. There were no main effects or interactions involving set as a factor.

Feature values show a clear U-shape pattern for loudness, valence and arousal, whereas tempo exhibits an inverted U-shape (Fig 5). Songs in the first and last positions of the albums exhibited the two highest valence (z-scores of 0.459 and –0.006, respectively) and loudness levels (0.36 and 0.006). Tempo was lowest in the first and last positions (–0.07 and –0.08) whereas arousal was stronger in the first position (0.26), but below average in the last one (–0.07).

To assess the statistical significance of the observed U-shaped relationships, a post-hoc ANOVA was performed to compare the residuals of a linear model with those of a quadratic model. Both models were constructed to explain the effect of position (ranging from 1 to 5) and feature type (i.e., valence, arousal, loudness, and tempo) on feature values. Results indicated that the model including a quadratic term for position $[F(5, 2594) = 9.172, p < .001]$ had a significantly better fit than the linear model $[F(5, 2594) = 6.05 p < .001]$, with lower residual sum of squares, and a higher adjusted explained variance ($RSS_{linear} = 2558.2$; $RSS_{quadratic} = 2534.2$; $R^2_{linear} = .009$; $R^2_{quadratic} = .018$.

Overall dissimilarities (Fig 6) between album positions were not different between control and experimental condition (P1/P2 = 3.15, P2/P3 = 3.05, P3/P4 = 3.1, P4/P5 = 3.06, with all $p$>.05). Similarly, tempo dissimilarities (P1/P2 = 69.71, P2/P3 = 71.94, P3/P4 = 70.77,

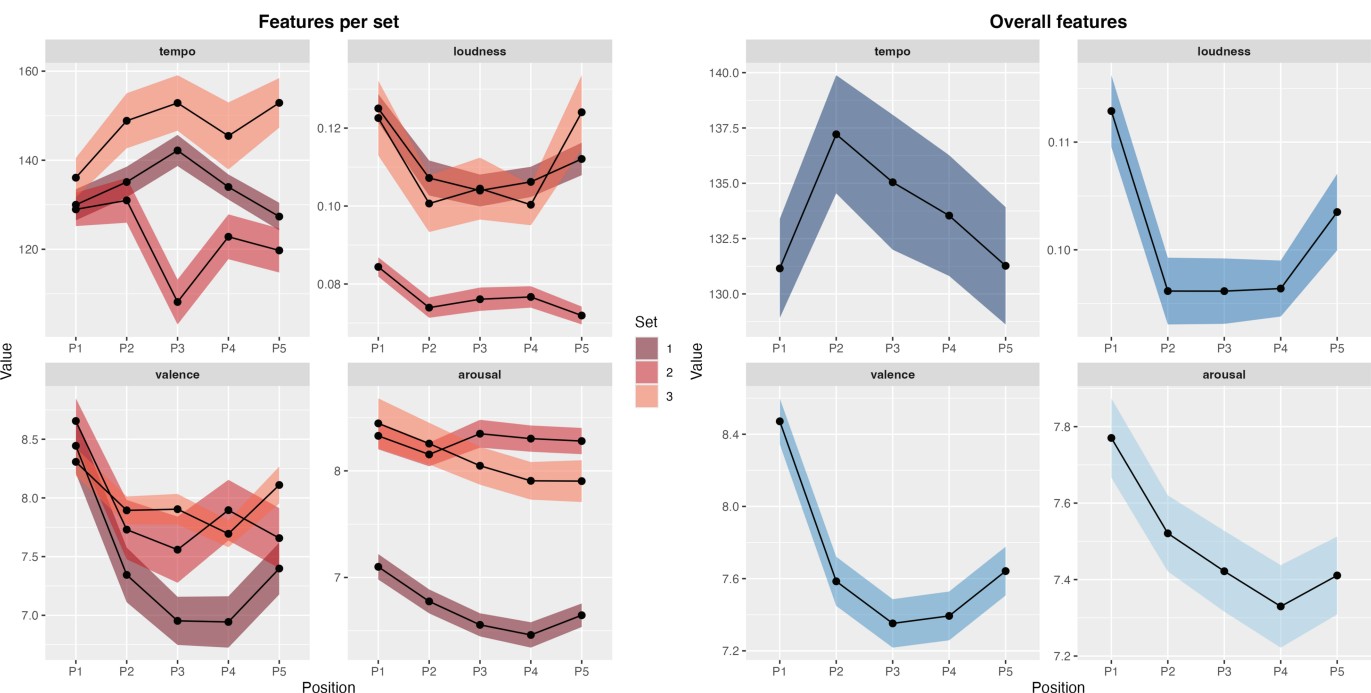

**Fig 5. Feature distributions.** Feature distribution per set and averaged across all sets. Shaded area represents the standard error of the mean for each position and feature.

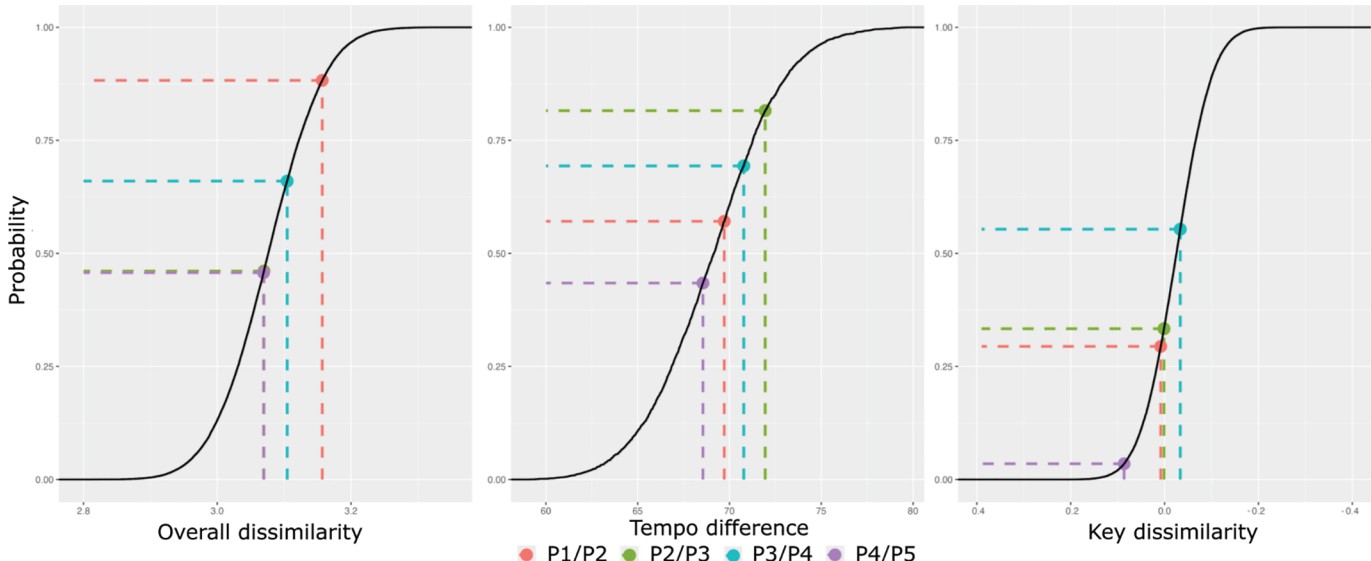

**Fig 6. Empirical cumulative distributions of track similarities.** Key dissimilarity, tempo dissimilarity and overall feature dissimilarity between pairs of consecutive tracks in the experimental condition (dashed lines), in relation to the Empirical Cumulative Distribution in the control condition. Values in the y-axis represent the probability of obtaining a given value or higher under the current ECDF.

P4/P5 = 68.55) did not differ from the random control distribution (M = 69.05, SD = 3.27, $p > .05$). Finally, p-values for key dissimilarity between consecutive album positions were statistically significant only for pair of tracks P4/P5 = 1.086 ($p < .05$), but not for the other ones (P1/P2 = 1.008, P2/P3 = 1.001, P3/P4 = 1.033; $p > .05$) when compared to the control condition (M = 1.02, SD = .06).

### 3.3 Survey

Results of the $\chi^2$ test were statistically significant for all but proposition D, indicating that responses for that proposition are relatively balanced between disagree, neutral and agree. If compared to the continuous responses for the same propositions (Fig 7) we see that, in fact, responses are shaped in 3 strong peaks, roughly centered at 25 (disagree), 50 (neutral) and 75 (agree).

All other $\chi^2$ statistics were significant at $p < .05$ (detailed statistics available as Supporting Information S3. Detailed survey statistics), which indicates that participants are biased towards one or two particular sides of the scale. The importance of track sequence in album production was the least controversial item of the survey, with most participants (84%) indicating their agreement with proposition (a) (Fig 7). Proposition B was also relatively uncontroversial, with participants agreeing that albums should not present very similar songs consecutively (63%).

Propositions (c), (d), (f), (g) and (h), were the most controversial ones, with many responses centered around 3 levels of the scale. With the exception of proposition (d), the distribution of responses for these propositions was still statistically significant, which indicates that, despite polarization, participants were biased towards at least one particular end of the scale.

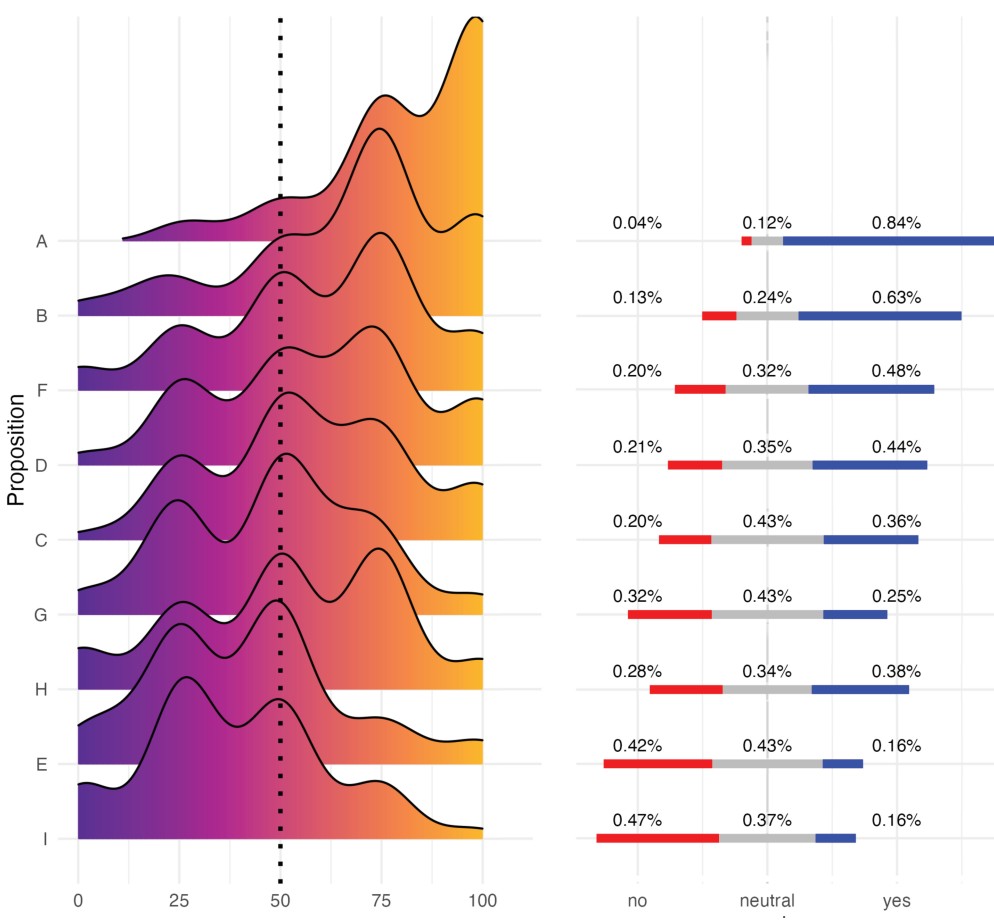

**Fig 7. Distribution of survey responses.** Density plots show the distribution of opinions throughout a continuous scale between completely agree and completely disagree. Bar plots show the proportion of respondents who agreed, disagreed, or were neutral about a given proposition.

Propositions (e) and (i) (Fig 7) received the highest frequencies of responses in the left end of the scale (i.e., disagree), indicating that musicians mostly disagree with the proposition that albums should end with the most striking and energetic songs, and that consecutive songs are to be as different as possible from each other, respectively.

Due to its exploratory nature, we refrain from conducting further hypothesis tests on individual propositions, and we consider that visualizations presented in Fig 7 provide enough information for us to discuss the results. All data and code utilized in this study are available from https://osf.io/69n3x (DOI 10.17605/OSF.IO/69N3X).

## 4 Discussion

Our findings demonstrate significant levels of agreement between music professionals regarding how tracks should be sequenced in an album. As measured by $PR$, particularly high levels of agreement were found for the first position of the albums. Results of the ANOVA suggest that the position within the album significantly influences the feature values, and this effect varies depending on the feature type. Furthermore, there was no effect of set, indicating that patterns were consistent irrespective of the songs and genres chosen for each set of tracks.

## 4.1 Album sequencing, emotional arcs and storytelling

**4.1.1 Literary works.** As shown in Fig 5, opening and closing tracks show high levels of valence and arousal. Following the circumplex model ([38]), this combination of features represents emotions such as excitement, delightfulness, and happiness. Middle tracks, on the other hand, show lower levels of valence and arousal, which corresponds to core emotions such as depression, boredom, and tiredness (for visual representations of the circumplex core emotions, see [39]).

With respect to valence, emotional arcs found in our albums bear a striking resemblance to those found in literary works. In their groundbreaking research, [40] performed computational sentiment analysis on a body of 1327 books from the Gutenberg Project (A), and found that 6 types of emotional arc generally prevail. These arcs correspond to the change in positivity (valence) of the narrative from start to finish, and have been categorized as follows:

1. Rags to riches—a steady rise from bad to good fortune
2. Riches to rags—a fall from good to bad, a tragedy
3. Icarus—a rise then a fall in fortune
4. Oedipus—a fall, a rise then a fall again
5. Cinderella—rise, fall, rise
6. Man in a hole—fall, rise

The Man in a hole arc, characterized by fall and a rise of positivity, was one of the most common arcs found by [40], accounting for 30% of all arcs in the literary works analyzed. Similar to literature, our findings suggest that album production favors the Man in a hole arc. This is true for overall trajectories of valence, but also for each individual set of tracks (Fig 5, left side).

It should be noted, however, that the textual features analyzed by [40] are essentially different from the audio features from our current study. It is still unclear, therefore, the extent to which similarity of emotional arcs are comparable from a perceptual standpoint. Furthermore, here we analyzed valence for aggregates of track sequences, whereas [40] analyzed emotional arcs followed by individual books. Still, even if other types of emotional arcs were also present in our sample, there is a strong prevalence of Man in a hole valence arcs within our album sequences, as revealed by our analysis of feature sequencing. Future work could still address the existence of alternative arcs within professional music albums.

**4.1.2 Song lyrics.** In a recent study, [41] analyzed the progression of storytelling elements in a large database of song lyrics. Specifically, the authors analyzed the distribution of staging, plot progression and cognitive tension across different segments of songs. Their findings revealed that lyrics tend to start with higher rates of words related to staging and cognitive tension, which gradually declines as the song progresses. Notably, cognitive tension peaks midway through the songs.

Again, there are major differences between our study and that of [41]. While here we focus on audio features and music albums, Alberhasky [41] investigate lyrics of individual songs. In fact, valence, arousal, loudness, and tempo can hardly be compared to staging, plot progression and cognitive tension. Still, our studies reveal strong similarities in the overall shapes of feature progressions. For instance, cognitive tension decreases throughout the song, similar to the way that valence and arousal tend to decrease throughout an album.

Whether or not these similarities are mere coincidences remains to be investigated. It is likely, however, as suggested by [12,41,41], that these patterns reflect some perceptual aspects that generally play a role when stimuli are presented sequentially, be it in audio features, lyrics, or literary plots. Future research may also expand on the work of [41], and analyze how

lyric-related features evolve throughout segments of an album or concert. In the next section, we raise hypotheses about the perceptual relevance of track sequences.

## 4.2 Why do musicians care about track sequencing?

So far we have provided empirical support for the idea that professional musicians see track sequencing as an important aspect of album production. The perceptual effect that different sequences of tracks may or may not have from the perspective of the listener, however, remains to be explored. It is worth noticing, in this respect, that while we found a significant effect of position in the album on feature values, small effect sizes raise questions about the practical relevance of these results. One could hypothesize, for instance, that track sequencing is an idiosyncrasy of music professionals, and that it is completely irrelevant to those who actually listen to the album.

In literary works [40], the choice of emotional arc has significant correlations with how successful the story is, as measured by the number of downloads that a particular book accumulates. For music, however, research suggests that randomly shuffling movements from Beethoven's compositions affected neither the pleasantness nor the emotional impact of these pieces [42]. These findings have been replicated with different methods, composers and music genres [43–47].

On the other hand, a significant body of research has shown that the perception of a given sound is dependent on the quality of the musical features that precede it. The renowned probe-tone paradigm demonstrates, for instance, that a single pitch can be perceived differently depending on the notes that come before it [48]. This is supported by other research paradigms, which generally show that musical context influences how individuals react to music both behaviorally and physiologically [49–52].

In light of auditory science, it seems unlikely that track sequencing has no impact on any facet of auditory perception. The studies [53–56] show, for instance, that variations in sound intensity, such as sudden and abrupt increases in Sound Pressure Level (SPL), can affect listener's attention, chills, and levels of arousal. The concept of loudness adaptation [57,58] reveals that the perception of intensity diminishes over time, even if the stimulus remains at a constant SPL. This shows that not only the immediate physical properties of the stimulus are capable of influencing how a sound is perceived, but also the context in which the stimulus is inserted. Future work on album sequencing carries the burden of unraveling what–if any–aspect of music perception is shaped by different sequences of tracks, whether inside or outside the context of an album.

## 4.3 Comparison with our previous study

Neto et al. [12] found a negative linear trend of valence, energy, loudness and tempo across album positions (Fig 1). This is partially in contrast with our current findings, which show that albums tend to start and end with lower levels of tempo (Fig 5). Similar to our large-scale Spotify analysis, however, valence, arousal and loudness were higher for opening tracks, although here these features followed a U-shaped curve (Fig 5). Such discrepancies could be attributed to at least two factors: (1) difference between MIRtoolbox and Spotify's feature extraction methods; (2) difference in number of albums analyzed.

We could hypothesize, for instance, that audio features computed by Spotify do not reflect the same dimensions of the features that are computed by MIRtoobox. Whereas the latter is an open-source and peer reviewed feature extraction tool, the former is proprietary method, which does not specify details of how high level features such as valence and energy are computed.

## 4.4 Comparison between survey responses and track sequences

Considering that this study is mainly inspired by anecdotal opinions about album sequencing, it is interesting to evaluate the extent to which explicit statements about song sequencing actually relate to the way that tracks are positioned on an album. However, as we have only checked for statistical significance of frequency distributions across different scale segments (i.e., agree, disagree and neutral), this section of our study should be interpreted as exploratory.

There was notable agreement between some propositions and sequencing data: participants believed that track sequencing is an important aspect of album production (proposition (a), Fig 7), a sentiment supported by the high levels of PR (Fig 4). With regards to audio features, most participants believed that albums should start with high-energy tracks (proposition (c), Fig 7), and our analysis showed that, in fact, albums often began with tracks exhibiting higher levels of arousal (Fig 5), but this effect did not appear to be very pronounced within individual sets. Moreover, participants mostly disagree that albums need to end with high energy levels (proposition (e), Fig 7), and this is in line with the arousal levels found in our feature analyses (Fig 5). Lastly, participants showed contrasting opinions about tonal similarity of consecutive tracks (proposition (h), Fig 7), and our key-similarity analysis showed that this feature was randomly distributed throughout all albums and sets, with the exception of one pair of tracks (Fig 7).

However, some opinions expressed in the survey diverged from what was observed in our sequencing data. While many participants felt that consecutive tracks should not be too similar, but also not maximally different from each other (propositions (b) and (i) respectively, 7), overall feature dissimilarity values were not statistically different between control and experimental condition (Fig 6). Even though the majority of participants said that tracks should be somewhat different with regards to tempo (proposition (f), Fig 7), tempo dissimilarities were also not statistically distinct from the random distribution (Fig 6).

In an overview, dissimilarity ratings between consecutive songs seemed to be secondary to overall arches of energy, valence and loudness. Even though participants had some opinions about the optimal contrast between pairs of tracks, these opinions generally did not reflect on the way that albums were sequenced. The comparisons provided here should, however, be considered in light of the fact that terms used in our propositions do not necessarily reflect the same musical feature computed in our analyses. It is likely, for instance, that the expression energetic song is understood as some musical quality that is not well captured by the way that MIRtoolbox computes arousal.

In Eerola et al.'s model [28], arousal is positively correlated with loudness, as well as with loudness variations–computed as the mean and standard deviation of the signal's root mean square, respectively–and negatively correlated with key clarity, which indicates how well the song conforms to one of the 12 keys in both major and minor modes. Valence, on the other hand, is negatively correlated with variations of loudness, and positively correlated with key clarity. In this model, both valence and arousal are also influenced by features that do not have direct interpretability, such as the entropy of the smoothed and collapsed spectrogram [28].

We do not know the extent to which these features, which have been shown to be relevant for the perception of emotion in music, can be linked to the terms that we employed in our survey. For a stronger analysis between survey opinions and sequencing regularities, future work in this area could apply more rigorous psychometric measurements, with satisfactory values of test-retest reliability and construct validity.

### 4.5 Limitations

**4.5.1 Positional Regularity as a Conservative measure.** Our proposed measure of *PR* is conservative in the sense that it reflects the agreement of musicians towards a specific track. Therefore, high levels of PR are only obtained if participants agree about the specific track which should be positioned in the beginning of an album, for instance.

It is the case, however, that each set of tracks in our samples provided individuals with more than one option of songs with a high value of valence, arousal, loudness and tempo. If tracks were not categorized by their names, but rather by their feature ranges (e.g., high, mid, low), we would be likely to see higher levels of *PR*.

In fact, the clear U-shape of valence in set 1 (Fig 5) suggests that participants were choosing higher values of valence for the final positions. Despite this agreement, PR was relatively low for *P*5 of the same set. Still, we consider that the feature analysis is enough to convey the levels of agreement that music professionals have towards the positioning of tracks according to their feature characteristics, and *PR* continues to be a coherent and feature-agnostic method to measure agreement about specific tracks.

**4.5.2 Number of tracks.** We chose to present participants only with albums that are composed of 5 tracks. Even though the sample of [12] comprised a significant number of albums with this exact length (6680 albums out of the 50000), and sequencing regularities were consistent throughout albums of different lengths, we do not know the extent to which our results would be generalizable to albums with distinct numbers of tracks. Future studies should address the effect that this variable may have in sequencing patterns.

**4.5.3 Sequencing and music taste: an alternative hypothesis.** A limitation of our work is the fact that we did not ask how much individuals liked each song. It could be hypothesized, for instance, that the most liked tracks in our sample were those positioned in the first and last positions of the album. Individuals could, in fact, be biased towards their music preferences, and therefore our sequencing regularities would no longer be attributable to the features of the track, but rather to an agreement about the quality of the songs. If that was the case, however, we would expect to see a wider variability in feature statistics throughout sequence positions. Considering previous literature on music taste, personal preferences span a wide range of music features between individuals [59].

Furthermore, this study partially replicates our previous results [12], which also suggests that published albums tend to have higher levels of energy in their opening sections (here reflected on valence and loudness values). Again, it is possible that these results also reflect a link between likable songs and high levels of loudness and valence.

While we did not explore the influence of musical genre, background, and preferences on album sequencing, our results suggest that sequencing regularities may have some level of independence from these factors, as PRs were significantly higher than chance across an heterogeneous sample of participants and musical genres, and position was found to be a statistically significant factor in our mixed-design ANOVA.

In face of these limitations and alternative hypotheses, we still consider that our results offer strong empirical evidence for the general hypotheses that (1) musicians consider track sequencing an important aspect of album production, and (2) that music-related features influence the way that professionals choose to organize their albums.

**4.5.4 Demographics and levels of musicianship.** In this study, we did not formulate a specific hypothesis regarding the potential influence of musicianship (e.g., years of experience, popularity, musical genre) or demographics (e.g., age, gender, nationality) on track sequencing regularities. However, it is possible that different musicians employ distinct sequencing

strategies. Future research could explore this further, either from our available dataset, or by conducting new studies that target specific musician and demographic populations.

## 5 Conclusion

Professional musicians consider track sequencing an important aspect of album productions. When asked to determine the order of tracks, participants show high levels of agreement about which position particular songs should occupy. Overall, tempo follows an inverted U-shape, whith faster songs being positioned in the middle of the album, whereas loudness, valence and arousal follow an inverted pattern. Our findings may be of relevance to music producers, streaming platforms, and listeners engaged in curating and/or generating music albums and playlists.

## Supporting information

**S1. Track information.** Details of the musical tracks used in the study, including album names, artist names, dataset sources, and track correspondence. Includes participant musical experience data summarized in tables.
(PDF)

**S2. Participants' musical experience.** Mean and standard deviation (in parentheses) of albums recorded, number of concerts performed, and years of experience as professional musicians across the three track sets.
(PDF)

**S3. Detailed survey statistics.** Chi-square and Dip-test statistics and their corresponding p-values for each survey proposition.
(PDF)

## Author contributions

**Conceptualization:** Pedro Neto, Geoff Luck, Petri Toiviainen.

**Data curation:** Pedro Neto.

**Formal analysis:** Pedro Neto, Martin Hartmann, Petri Toiviainen.

**Funding acquisition:** Geoff Luck, Petri Toiviainen.

**Investigation:** Pedro Neto, Geoff Luck, Petri Toiviainen.

**Methodology:** Pedro Neto, Martin Hartmann, Geoff Luck, Petri Toiviainen.

**Project administration:** Geoff Luck, Petri Toiviainen.

**Resources:** Pedro Neto, Petri Toiviainen.

**Software:** Pedro Neto.

**Supervision:** Martin Hartmann, Geoff Luck, Petri Toiviainen.

**Validation:** Pedro Neto, Martin Hartmann.

**Visualization:** Pedro Neto, Petri Toiviainen.

**Writing – original draft:** Pedro Neto, Martin Hartmann, Geoff Luck, Petri Toiviainen.

**Writing – review & editing:** Pedro Neto, Martin Hartmann, Geoff Luck, Petri Toiviainen.

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
