## [Editor Report · Decision Letter 0]

9 Jan 2025

PONE-D-24-54145An album is a story: feature arcs in sequences of tracksPLOS ONE

Dear Dr. Neto,

Thank you for submitting your manuscript to PLOS ONE. After careful consideration, we feel that it has merit but does not fully meet PLOS ONE’s publication criteria as it currently stands. Therefore, we invite you to submit a revised version of the manuscript that addresses the points raised during the review process.

We look forward to receiving your revised manuscript.

Kind regards,

Seung-Goo Kim, Ph.D.

Academic Editor

PLOS ONE

Journal Requirements:

“This research was funded by the Research Council of Finland, under the Center of Excellence in Music, Mind, Body and Brain (Project number: 346210).”

4. Please note that your Data Availability Statement is currently missing the repository name and/or the DOI/accession number of each dataset OR a direct link to access each database. If your manuscript is accepted for publication, you will be asked to provide these details on a very short timeline. We therefore suggest that you provide this information now, though we will not hold up the peer review process if you are unable.

Additional Editor Comments:

One of the PLOS's core values is transparency. As an academic editor, it is my responsibility to ensure that all authors of the articles published in PLOS ONE have made the data underlying their findings fully available. Please take a look at our criteria for publication (https://journals.plos.org/plosone/s/criteria-for-publication).

The current submission does not include where the data can be retrieved. The authors' input is simply a copy-and-paste of the sample text: 'The data underlying the results presented in the study are available from XXX'. Please see our resource (https://journals.plos.org/plosone/s/data-availability) to understand what constitutes the 'minimal data set' and ideas about sharing data.

Once all requirements for submission are met, then I will consider sending it out to reviewers for further assessment.

---

## [Author Response · Author response to Decision Letter 1]

28 Jan 2025

We have made all the requested changes and additions to our submission. Detailed responses to each point raised by the editor are available in the attached file "Response to reviewers".

---

## [Decision Letter · Decision Letter 1]

20 Feb 2025

PONE-D-24-54145R1An album is a story: feature arcs in sequences of tracksPLOS ONE

Dear Dr. Neto,

Thank you for submitting your manuscript to PLOS ONE. After careful consideration, we feel that it has merit but does not fully meet PLOS ONE’s publication criteria as it currently stands. Therefore, we invite you to submit a revised version of the manuscript that addresses the points raised during the review process.

Please note that all reviewers evaluated the study highy positive while raising constructive points that would substantiate the current manusscript. In addition, the Academic Editor also added methodological points. All raised points must be addressed in a point-by-point manner in the rebuttal.

We look forward to receiving your revised manuscript.

Kind regards,

Seung-Goo Kim, Ph.D.

Academic Editor

PLOS ONE

Additional Editor Comments:

In addition to the reviewers' comments, I would like to add a few methodological points.

1. [line 169] ANOVA only tests for the inequality of group means and does not support the findings of U or inverted-U patterns. Given that this is one of the main conclusions, it needs to be explicitly tested. To test the U or inverted U patterns, a quadratic fit can be compared with a linear fit using an F-test (e.g., general linear model).

2. [line 157] It needs to be clarified that "valence" and "arousal" are only estimates (or predictions) from the MIR toolbox and not human-rated measures. The cited studies are not the validation of the MIR toolbox estimates (which should have been presented in this context), but the investigation of the psychological constructs ("valence" and "arousal") that were self-reported by human participants. However, readers may not know how good (or poor) the MIR toolbox estimates are in predicting human ratings. It must be clarified that the "valence" and "arousal" are not human ratings but models' predictions. Also, if possible, provide validation studies that support the validity of the MIR toolbox in terms of those features.

3. [Equation 2] The averaged l2-norm of the first-order difference of the feature vectors (or scalars) was defined as an "overall dissimilarity" measure and tested against the control condition (i.e., H0: random changes) [Figure 6]. But, it seems that the changes between particular positions were reported in the text (e.g., "P1/P2 = 3.15" [line 233]), not the "overall dissimilarity" measure, which should have been averaged across all transitions. This needs to be consistently reported.

4. [line 183] When defining a distance metric based on Pearson correlation, 1-r is more conventional for its range will be positive [0, 2].

5. [line 193] It is unclear why the authors want to test "whether or not participants leaned toward one or more segments of the scale" [line 203]. As shown in [line 97], the survey was carried out to find what are the factors that the musicians would say. For example, "Do they think a good sequence is important or not". Now the authors are answering a different question: "Do they think sequence is uniformly important and unimportant?" But why do you want to know this instead of the first question? Is this a sign of p-hacking?

- Generally binning needs to be better justified. The non-Gaussianity of the ordinal measure such as the Likert scale can be still dealt with by other non-parametric alternative approaches such as the rank-sum test.

- Even if the binning can be justified, why unequal segments as [45%, 10%, 45%] instead of not [33%, 33%, 33%]?

Reviewers' comments:

Reviewer's Responses to Questions

**Comments to the Author**

1. If the authors have adequately addressed your comments raised in a previous round of review and you feel that this manuscript is now acceptable for publication, you may indicate that here to bypass the “Comments to the Author” section, enter your conflict of interest statement in the “Confidential to Editor” section, and submit your "Accept" recommendation.

Reviewer #1: (No Response)

Reviewer #2: (No Response)

Reviewer #3: (No Response)

2. Is the manuscript technically sound, and do the data support the conclusions?

Reviewer #1: Yes

Reviewer #2: Yes

Reviewer #3: Yes

3. Has the statistical analysis been performed appropriately and rigorously? 

Reviewer #1: Yes

Reviewer #2: Yes

Reviewer #3: Yes

4. Have the authors made all data underlying the findings in their manuscript fully available?

Reviewer #1: No

Reviewer #2: Yes

Reviewer #3: (No Response)

5. Is the manuscript presented in an intelligible fashion and written in standard English?

Reviewer #1: Yes

Reviewer #2: Yes

Reviewer #3: Yes

6. Review Comments to the Author

Reviewer #1: Dear Authors,

I had the opportunity to review your manuscript, “An album is a story: Feature arcs in sequences of tracks”. The manuscript empirically investigates whether there is truth to the claim that musicians implement consistent sequencing principles when arranging tracks in a musical album. The manuscript reveals statistically relevant patterns of track sequencing, suggesting that this process is based on both audio features and the position of a track within an album.

Overall, my impression of the manuscript is positive. However, I have noted down a few comments which I invite you to further consider. I have explained these comments in a point-by-point format, below:

Introduction

---

[1] I find the topic of the manuscript quite interesting, and while it was easy for me to understand why this topic is relevant from a theoretical standpoint (the lack of systematic investigations into the matter; p. 2, lines 32-33), I struggled to see the practical relevance of it. Would understanding specific sequencing principles help an album’s total sales or streams? A similar practical point is mentioned in the Discussion with regards to story arcs in literary works (p. 9, lines 328-329), but not in the Introduction. I feel that the manuscript would benefit from a short elaboration of why understanding album sequencing principles would be practically relevant.

Methods

---

[2] The standard deviations for participants’ average concerts and recorded albums suggest that the variance within these two demographics is relatively high (p. 2, lines 54-56). Were there any differences in sequencing principles between participants who were more of less experienced in album releases or concerts?

[3] I felt that the Methods section missed a few small, but relevant, details. For instance, while I can understand the reasoning behind using jazz and classical music to control for potential instrumentation effects in sequencing practices, I missed why it was necessary to come up with 3 sets of 5 songs in the first place. What was the purpose of comparing the 3 sets?

[4] Next, in the selection of stimuli for set 1, what made a 60 second segment “representative” of a particular track (p. 3, lines 87-88)? Compositionally, e.g., in terms of musical features, were these 60 second segments similar to those in sets 2 and 3?

[5] Several measures of dissimilarity between consecutive tracks were calculated (p. 5, lines 169-191) based on different (combinations) of audio features. However, their purpose was not immediately clear to me. Could you please elaborate what information these dissimilarity metrics add beyond the initial ANOVA of feature values?

Furthermore, with regards to the computation of these dissimilarity measures, why was overall audio feature and tempo dissimilarity calculated via Euclidian distance, while key strength profile (dis)similarity was calculated via Pearson correlations? I feel that you must have had a clear reason for doing this, and it would be helpful if this reasoning were further explained.

Results

---

[6] The manuscript reports a statistically significant interaction effect between position and feature (p. 7, lines 229-230). Were any post-hoc tests conducted to further examine this statistically significant interaction?

Discussion

---

[7] Perhaps this is a bit of a layman’s perspective on the matter, but I feel that songs are rarely experienced as individual features, rather a combination multiple. Thus, while the results and subsequent discussion points highlight the role of individual audio features in track sequencing principles, I wonder whether musicians deliberately consider individual features as opposed to a combination of several in their sequencing choices. Is this something that can be inferred from the data obtained for the manuscript, and, if so, can the Discussion be expanded to address this?

Minor suggestions

---

[8] This is a slightly nit-picky comment, but if recruitment was done in three separate stages (p. 4, lines 115-116), one per set of songs, is it still accurate to say that participants were randomly assigned to each set of songs (p. 3, line 92)? Here too, were there any pronounced differences in demographics between participants at each stage?

[9] For consistency, I suggest changing the “Survey” header (p. 7, line 240) to “Questionnaire” (p. 3, line 97; or the other way around).

Reviewer #2: This experiment looks at whether there is a particular sequence to how tracks unfold on a musical album. The authors had 130 participants sequence various album sets consisting of 5 tracks. The participants also answered a survey about their views on issues related to sequencing. Overall, the participants showed broad agreement on the placement/sequence of the songs. When analyzing the musical features of the songs, the researchers noticed that tempo follows an inverted U Shape while valence, arousal, and loudness exhibited the opposite pattern. The authors note the general trajectory of album songs is like what is seen in the literary domain, and I found this connection to other art forms quite intriguing.

In general, the paper is well-written, and the experiment seems well-designed. I am unable to speak much to the specific statistical analyses used, but based on the explanation, these approaches seem valid. The paper was an interesting read, and I only have a couple of minor comments, although these do not detract from the overall paper.

Participants: Could more details be given about these professional musicians and music producers who were recruited? For example, what type of musical training did they have? I like that the track sequencing focused on little-known instrumental jazz and classical music, however, I would like to know more about these participants. Do they have a lot of experience with jazz or classical music? Do we think this may alter how they sequence the tracks?

I understand using 5 tracks for this experiment as that seems like a reasonable amount for participants to handle sequencing. However, from an ecological validity standpoint, most albums have more than that, say 10-20 tracks. How did the authors settle on 5 tracks per set?

I really liked using the survey to investigate agreement with statements about perspectives on sequencing. I think future studies might look at WHY participants feel this way. Do we think there is something “universal” or general about auditory sequencing? In other words, is there a certain “story” that is told when sequencing tracks? Also, I imagine results may vary a lot if these were songs with lyrics. I think some of this could be mentioned, even if it’s exploratory, in the discussion to ground why it’s important to think about album sequencing (i.e. the significance).

Reviewer #3: Dear Authors,

Thank you for submitting your article to PLOS ONE. I particularly enjoyed reading about a topic I found very interesting.

The manuscript presents a well-structured and methodologically sound study, addressing an interesting intersection between music perception and auditory science. The use of professional musicians and the inclusion of different musical genres enhance the ecological validity of the findings. The authors provide clear statistical analyses, supported by ANOVA and bootstrapping approaches, ensuring robust results. However, some aspects could benefit from further clarification, particularly regarding the interpretation of feature dissimilarity and the potential impact of individual track preferences. Additionally, while the authors acknowledge limitations, the discussion of how sequencing preferences might vary across musical genres could be expanded.

Please find some comments below:

1. Abstract:

Clarify how the “inverted U-shape” of tempo and the opposite pattern of valence, arousal, and loudness were measured. It might be helpful to mention the use of MIRtoolbox here to strengthen the methodological transparency.

2. Introduction (Page 2, Line 40-44):

The introduction mentions a previous study involving 51,010 published albums. It would be useful to briefly explain whether the current study expands on those findings or addresses a different aspect of track sequencing.

3. Materials and Methods (Page 3, Line 62-72):

The rationale for selecting only jazz and classical genres is well-explained, but the authors might consider addressing whether findings could generalize to more popular or rhythmically consistent genres, such as pop, rock, or EDM.

4. Sequencing Analysis (Page 4, Line 130-140):

The calculation of Positional Regularity (PR) is detailed, but it might be clearer to provide an example illustrating how PR reflects consensus among participants.

5. Results (Page 7, Line 221-230):

While the ANOVA results are statistically significant, the η² values suggest small effect sizes. It would be helpful to discuss whether these effects, while significant, are practically meaningful in real-world album production.

6. Discussion (Page 8, Line 272-283):

The comparison between emotional arcs in album sequencing and literary works is compelling but somewhat speculative. Consider adding a disclaimer emphasizing that while the patterns are similar, the underlying cognitive mechanisms might differ.

7. Conclusion (Page 12, Line 464-469):

The conclusion effectively summarizes the findings but could benefit from emphasizing the study’s implications for music producers and platforms like Spotify or Apple Music, where track sequencing might influence user experience.

Finally: Please carefully check the spacing within parentheses and remove unnecessary spaces (e.g., p.1, lines 33 and 49, etc.).

7. PLOS authors have the option to publish the peer review history of their article (what does this mean?). If published, this will include your full peer review and any attached files.

Reviewer #1: No

Reviewer #2: **Yes: **Karen Chan Barrett

Reviewer #3: No

---

## [Author Response · Author response to Decision Letter 2]

24 Mar 2025

Dear Editor and Reviewers,

We were very pleased with your comments, and I believe they helped us make significant improvements to our paper. In the document "Response to reviewers" you will find our answers to each one of the issues that you raised.

A practical note: The page and line numbers in your references to our manuscript did not align with our local document, likely due to differences in LaTeX rendering. To ensure clarity, we refer to Section. Subsection. Paragraph number in our responses, all highlighted in yellow.

Best regards,

P.N.

---

## [Decision Letter · Decision Letter 2]

15 Apr 2025

An album is a story: feature arcs in sequences of tracks

PONE-D-24-54145R2

Dear Dr. Neto,

We’re pleased to inform you that your manuscript has been judged scientifically suitable for publication and will be formally accepted for publication once it meets all outstanding technical requirements.

Kind regards,

Seung-Goo Kim, Ph.D.

Academic Editor

PLOS ONE

Additional Editor Comments (optional):

Reviewers' comments:

Reviewer's Responses to Questions

**Comments to the Author**

1. If the authors have adequately addressed your comments raised in a previous round of review and you feel that this manuscript is now acceptable for publication, you may indicate that here to bypass the “Comments to the Author” section, enter your conflict of interest statement in the “Confidential to Editor” section, and submit your "Accept" recommendation.

Reviewer #1: All comments have been addressed

Reviewer #2: All comments have been addressed

Reviewer #3: All comments have been addressed

2. Is the manuscript technically sound, and do the data support the conclusions?

Reviewer #1: Yes

Reviewer #2: Yes

Reviewer #3: Yes

3. Has the statistical analysis been performed appropriately and rigorously? 

Reviewer #1: Yes

Reviewer #2: Yes

Reviewer #3: Yes

4. Have the authors made all data underlying the findings in their manuscript fully available?

Reviewer #1: Yes

Reviewer #2: Yes

Reviewer #3: Yes

5. Is the manuscript presented in an intelligible fashion and written in standard English?

Reviewer #1: Yes

Reviewer #2: Yes

Reviewer #3: Yes

6. Review Comments to the Author

Reviewer #1: (No Response)

Reviewer #2: I have read the revised manuscript and believe that the critiques of all three reviewers were aimed at ensuring we understood the significance/application of this study, as well as situating this study among existing information and previous research. The authors responded effectively to each of my critiques.

I also like the newly-added text in the manuscript about 1) the previous study in the introduction so that we now know why certain decisions were made in this study since it builds upon that one, 2) explanation of High vs. Low PR scores as well details about the feature analysis, 3) why they analyzed the survey data the way they did and 4) addressing reviewer critiques through the discussion and limitations sections. I think the manuscript is much improved with these additions, and I am satisfied with the resubmission.

Reviewer #3: Dear Authors,

Thank you for your thorough and thoughtful revisions. After reviewing the revised manuscript, I am pleased to note that you have addressed all the concerns raised. The improvements have enhanced the clarity and quality of the paper. I have no further comments and recommend the manuscript for publication.

Best regards,

7. PLOS authors have the option to publish the peer review history of their article (what does this mean?). If published, this will include your full peer review and any attached files.

Reviewer #1: No

Reviewer #2: **Yes: **Karen Chan Barrett

Reviewer #3: No

---

## [Editor Report · Acceptance letter]

PONE-D-24-54145R2

PLOS ONE

Dear Dr. Neto,

I'm pleased to inform you that your manuscript has been deemed suitable for publication in PLOS ONE. Congratulations! Your manuscript is now being handed over to our production team.

Kind regards,

on behalf of

Dr. Seung-Goo Kim

Academic Editor

PLOS ONE